# Clinically Relevant Oxygraphic Assay to Assess Mitochondrial Energy Metabolism in Acute Myeloid Leukemia Patients

**DOI:** 10.3390/cancers13246353

**Published:** 2021-12-17

**Authors:** Quentin Fovez, William Laine, Laure Goursaud, Celine Berthon, Nicolas Germain, Claire Degand, Jean-Emmanuel Sarry, Bruno Quesnel, Philippe Marchetti, Jerome Kluza

**Affiliations:** 1Institut pour la Recherche sur le Cancer de Lille, Univ. Lille, CNRS, Inserm, CHU Lille, UMR9020-UMR-S 1277-Canther-Cancer Heterogeneity, Plasticity and Resistance to Therapies, F-59000 Lille, France; quentin.fovez@inserm.fr (Q.F.); william.laine@univ-lille.fr (W.L.); laure.goursaud@chru-lille.fr (L.G.); nicolas.germain@inserm.fr (N.G.); claire.degand@inserm.fr (C.D.); bruno.quesnel@chru-lille.fr (B.Q.); philippe.marchetti@inserm.fr (P.M.); 2Hematology Department, CHU Lille, F-59000 Lille, France; celine.berthon@inserm.fr; 3Centre de Bio-Pathologie, Banque de Tissus, CHU Lille, F-59000 Lille, France; 4Centre National de la Recherche Scientifique, Centre de Recherches en Cancérologie de Toulouse, Institut National de la Santé et de la Recherche Médicale, Université de Toulouse, 31100 Toulouse, France; jean-emmanuel.sarry@inserm.fr

**Keywords:** energy metabolism, leukemia, resistance, functional biomarker, uncoupling respiration, OCR, XFe24 Seahorse, XFe96 Seahorse

## Abstract

**Simple Summary:**

AML mitochondrial oxidative phosphorylation has recently been identified as a biological property that influences the response to antitumor therapy. In the present study, we propose a standardized protocol to measure mitochondrial metabolic organization in patient blasts (from the blood or bone marrow) using XFe24 or XFe96 Seahorse. Monitoring mitochondrial oxygen consumption of blasts could improve the prediction of drug response in AML patients, especially in clinical trials.

**Abstract:**

Resistant acute myeloid leukemia (AML) exhibits mitochondrial energy metabolism changes compared to newly diagnosed AML. This phenotype is often observed by evaluating the mitochondrial oxygen consumption of blasts, but most of the oximetry protocols were established from leukemia cell lines without validation on primary leukemia cells. Moreover, the cultures and storage conditions of blasts freshly extracted from patient blood or bone marrow cause stress, which must be evaluated before determining oxidative phosphorylation (OXPHOS). Herein, we evaluated different conditions to measure the oxygen consumption of blasts using extracellular flow analyzers. We first determined the minimum number of blasts required to measure OXPHOS. Next, we compared the OXPHOS of blasts cultured for 3 h and 18 h after collection and found that to maintain metabolic organization for 18 h, cytokine supplementation is necessary. Cytokines are also needed when measuring OXPHOS in cryopreserved, thawed and recultured blasts. Next, the concentrations of respiratory chain inhibitors and uncoupler FCCP were established. We found that the FCCP concentration required to reach the maximal respiration of blasts varied depending on the patient sample analyzed. These protocols provided can be used in future clinical studies to evaluate OXPHOS as a biomarker and assess the efficacy of treatments targeting mitochondria.

## 1. Introduction

Acute myeloid leukemia (AML) is a malignant myeloid disease characterized by the loss of blast differentiation and clonal amplification in the peripheral blood and bone marrow. Over the past decade, improvements in AML diagnosis and advances in therapeutic approaches have improved the outlook for patients. However, despite this considerable progress, the five-year overall survival rate for AML is still 24% [1]. To date, the diagnosis and management of hematologic malignancies are primarily based on the cytogenetic and molecular characteristics of leukemic blasts.

European LeukemiaNet (ELN-2017) recommendations have determined the profiles of coexisting and exclusive mutations in blasts to provide prognostic information and allow for the stratification of patients into three risk categories (favorable, intermediate or adverse) [2]. The ELN classification is used to guide postremission treatment; notably, hematopoietic allogeneic transplant does not seem necessary for patients in complete first remission, whereas this therapy is strongly recommended for patients with adverse risk [3]. However, further refinements of the ELN-2017 classification are still under investigation [4]. Therefore, the evaluation of new biomarkers is needed to complete the ELN classification, and blast energy metabolism could be a potential candidate [5,6,7].

AML metabolism, in particular the energy functions of the mitochondria of leukemic blasts, has recently been identified as a biological property that influences the response to antitumor therapy. Pharmacological inhibitors of mitochondrial oxidative phosphorylation (OXPHOS) (e.g., oligomycin [8], CB839 [9] and other drugs such as tigecycline [10]) increase the AML cell death in vitro and promote the eradication of leukemic stem cells. In other hematopoietic diseases, such as multiple myeloma, electron transport chain activity is also a predictor of venetoclax sensitivity [11]. Moreover, molecules targeting mitochondria (e.g., complex I inhibitors) are currently under preclinical investigation or in phase I clinical trials for the treatment of AML [12]. Since mitochondrial metabolic activity supports AML resistance, mitochondrial oxygen consumption could be considered as a potential biomarker to predict the response to treatment. However, robust and efficient clinical markers of OXPHOS in leukemic blasts are still under investigation.

In particular, methods for measuring mitochondrial oxygen consumption have been developed in recent years. Based on the quantification of O_2_ dissolved in aqueous solution, polarographic oxygen sensors (Clark electrodes and others) have been widely used [13]. However, fluorometers that simultaneously assess OXPHOS and glycolytic activity (XFe24 and XFe96 Seahorse) are other alternatives to quantify the oxygen consumption of various cell models, including AML [14].

The oxygen consumption of patient blasts has been evaluated in several studies, but the different methodologies used (number of cells, concentration of inhibitors or culture conditions) do not allow comparison of the data obtained from different reports. In the present study, we propose a standardized protocol to measure mitochondrial metabolic organization in patient blasts using Seahorse XFe24 or XFe96. All tests were performed with blasts taken from the blood or bone marrow of patients with AML. The protocols were designed for clinical use at the time of AML diagnosis or relapse.

## 2. Materials and Methods

### 2.1. Chemicals

Oligomycin A, carbonyl cyanide p-trifluoromethoxyphenylhydrazone (FCCP), antimycin A, rotenone, BAM15, were purchased from Sigma-Aldrich (St-Louis, MO, USA). Stock solutions (10 mm or 2.5 mm) of each compound were prepared in sterile dimethyl sulfoxide (DMSO) and stored at −20 °C. All XFe24 and XFe96 plates were purchased from Agilent Technologies (Santa Clara, CA, USA).

### 2.2. Blast Collection and Cell Culture Conditions

AML samples (Table 1) were obtained from individuals recruited from the Department of Hematology (Lille CHU, France) with informed consent in accordance with the Declaration of Helsinki and with the approval of the Institutional Ethical Committee (CPP Lille). Blood or bone marrow samples were diluted twofold with PBS and gently placed on top of Pancoll human (d = 1.077 g∙mL^−1^) (PAN-Biotech, Aidenbach, Germany,) by running the blood against the wall of the tube. Samples were then centrifuged for 30 min (400 g–Acc.1 Dec.1). Mononuclear cells were then collected and washed twice in PBS followed by centrifugation for 5 min (300 g–Acc.9 Dec.9). If needed, the red blood cells were lysed with osmotic buffer (0.1 M KHCO_3_, 1.55 M NH_4_Cl and 0.037 g of Na_2_EDTA). After isolation, the cell number and cell viability were assessed using a cell counter (Z2 Coulter Counter, Beckman Coulter Inc., Hialeah, FL, USA) and a viability test (LUNA-FL™ Dual Fluorescence Cell Counter, Logos Biosystems, Inc., Gyeonggi-do, Korea) (trypan blue or propidium iodide).

Mononuclear cells were frozen in cryopreservation medium (90% fetal bovine serum (Gibco by Life Technologies Corp., Grand Island, NY, USA) + 10% DMSO (Sigma-Aldrich Corp, St Louis, MO, USA). Cryogenic tubes were placed in freezing containers containing isopropanol and stored at −80 °C for two to three days. The cryogenic tubes were then placed in a nitrogen tank for long-term storage. Samples were thawed in RPMI medium (Gibco) supplemented with 10% fetal bovine serum (Gibco), 50 U∙mL^−1^ penicillin, 50 mg∙mL^−1^ streptomycin and 100 µg∙mL^−1^ DNase I 1X (Sigma). Mononuclear cells were cultured at 37 °C with 5% CO_2_ in RPMI medium (Gibco) supplemented with 10% fetal bovine serum (Gibco), 50 U∙mL^−1^ penicillin, 50 mg∙mL^−1^ streptomycin. RPMI medium has been optionally supplemented with a cytokines cocktail containing 10 ng∙mL^−1^ interleukins (IL-3, IL-6, and IL-7), granulocyte colony-stimulating factor (G-CSF), 50 ng×mL^−1^ FMS-like tyrosine kinase 3 ligand (FLT3L), 5 ng×mL^−1^ granulocyte macrophage colony-stimulating factor (GM-CSF) and 25 ng∙mL^−1^ stem cell factor (SCF) (PeproTech, East Windsor, NJ, USA).

### 2.3. Oxygen Consumption Rate (OCR) Measurements

OCRs were measured using a Seahorse XFe24 or XFe96 analyzer (Seahorse Bioscience, Billerica, MA, USA). Seahorse XFe24 and XFe96 microplates were precoated the day of the experiment for at least 15 min with 3 µg∙cm^−2^ Corning^TM^ Cell-Tak solution (Fisher Scientific): 35 µL (XFe24) or 20 µL (XFe96). After the coating process, the microplates were rinsed with sterile water. Cells were suspended in OXPHOS medium containing DMEM (D5030, Sigma-Aldrich) with L-glutamine (2 mm), glucose (10 mm) and pyruvate (1 mm) and seeded in XFe24 cell plates (100 µL∙well^−1^) or XFe96 cell plates (50 µL∙well^−1^). Cells were left to adhere by two successive centrifugations at low speed (160 g for 1 min), and each microplate was then left to stabilize for at least 20 min at 37 °C in a CO_2_-free incubator. Next, 400 µL (XFe24) or 100 µL (XFe96) of warm OXPHOS medium was added to each well. Before each OCR measurement, calibration and equilibration of the Seahorse analyzers were performed. Each port was loaded with 75 µL (XFe24) or 20 µL (XFe96) of inhibitor diluted in OXPHOS medium. The OCR was assessed at baseline and after injection of each of the following molecules: oligomycin A, FCCP (or BAM15), rotenone and antimycin A. All settings are available in Table 2 and Appendix A.

### 2.4. High-Content Imaging

Imaging was carried out using a Cytation 1 Cell Imaging Multi-Mode Reader (BioTek Instruments, Winooski, VT, USA). The environment was controlled at 37 °C. Hoechst 33342 (35.5 µM final concentration) was imaged using a 365 nm LED in combination with an EX 377/50 EM 447/60 filter cube. Image analysis was performed using Gen5 software (BioTek).

### 2.5. Cytofluorometric Analysis

Cell viability and mitochondrial membrane potential (ΔΨM) were assessed by Annexin V-APC (0.45 µg per sample, 10 min, room temperature (RT) (Biolegend, San Diego, CA, USA), SYTOX blue (1 µM, 10 min, RT; Thermo Fisher Scientific, Inc., Waltham, MA, USA) and TMRM (100 nm, 30 min, 37 °C; Thermo Fisher Scientific) staining. Fluorescence intensity following cellular staining were analyzed with a FACS LSR Fortessa X20 (Becton Dickinson, Franklin Lakes, NJ, USA).

### 2.6. Statistical Analysis

All data are represented as the means ± SD or ± SEM. One-way or two-way analysis of variance (ANOVA) followed by Dunnett’s or Sidak post hoc test were used to compare mean values between multiple groups. Statistical analysis was performed using Prism version 6.0f (GraphPad Software, La Jolla, CA, USA). *p* values < 0.05 were considered statistically significant.

## 3. Results

### 3.1. Minimum Number of Blasts Required to Measure OXPHOS

To establish a standardized protocol to assess blast OXPHOS with Seahorse XFe24 or XFe96, all experiments were performed with blasts isolated from the blood or bone marrow of AML patients (Table 1). The blood of AML patients collected at the time of diagnosis usually contains a high number of blasts. Mononuclear cells were isolated from the patient’s peripheral blood by density gradient centrifugation on a Pancoll Human, cultured in full RPMI medium and washed and resuspended in OXPHOS medium before oxygen consumption evaluation. Because leukemia cells are nonadherent, they should be immobilized at the bottom of the XFe24 or XFe96 Seahorse microplates. Therefore, Cell-Tak coated plates were prepared before analysis.

First, we determined the appropriate number of blasts to seed in each well (the area per well in XFe24 plates is 28.3 mm^2^) to completely cover the wells without cell overlap. AML blasts were seeded at a density between 0.125 × 10^6^ and 1 × 10^6^ cells in XFe24 plates (Figure 1A), and brightfield images were immediately taken from several representative wells (Figure 1A, upper panel). The images showed cell overlay with 0.5 × 10^6^ cells. To confirm these results, the blasts were stained with Hoechst, and the nuclei were visualized using a fluorescence cell imager (Cytation I) (Figure 1A, middle panel). With 0.5 × 10^6^ cells, the cell overlay increased the blue background fluorescence. Under these conditions, cell counting cannot be correctly performed with an automatic detection method based on nucleus segmentation (Cell Imaging^®^) (Figure 1A, bottom panel). The same experiments were carried out in XFe96 Seahorse microplates (area of 11.4 mm^2^ per well), and cell overlay was observed with 1.8 × 10^5^ cells (Appendix A).

Next, real-time OCR measurements were taken by isolating a small volume (7 µL from the XFe24 plates or 2 µL from the XFe96 plates) from a transient microchamber of medium above the blasts attached to the microplate wells. The consumption of cellular oxygen causes a rapid decrease in the dissolved oxygen concentration compared to wells containing only medium (background) (Figure 1B or Appendix A). When the measurement was complete, the cartridge was lifted, which allowed a large volume of medium to restore the oxygen values in the medium surrounding the cells to the baseline. All parameters, including the number of microchamber open/close cycles, waiting time and measurement time, are summarized in Table 2 for the XFe24 Seahorse (and in Appendix A for the XFe96 Seahorse). Using these parameters, a significant decrease in oxygen concentration was observed during chamber closure (t_0 min_, t_7 min_, t_14 min_) in the XFe24 plates with 0.125 × 10^6^ cells, which became more pronounced with 0.25 × 10^6^ cells. With the XFe96 Seahorse, a significant decrease in oxygen concentration was observed with 0.9 × 10^5^ cells (Appendix A). In conclusion, to avoid blast overlay in the wells and detect significant variations in oxygen concentration during chamber closure, we decided to seed 0.25 × 10^6^ cells in the XFe24 plate and 0.9 × 10^5^ cells in the XFe96 plate for all the next experiments.

### 3.2. Sequential Injections of Inhibitors to Measure OXPHOS

The total amount of oxygen consumed by AML mitochondria represents only a partial picture of the OXPHOS process, so most protocols use sequential injections of inhibitors to measure the different states of mitochondrial oxygen consumption (Figure 1C, Appendix A or Appendix A). During the first three measurements, the basal OCR of the blasts was determined (from t_0 min_ to t_15 min_). This result represents the sum of all of the oxidative processes consuming O_2_, such as the mitochondrial activity of cytochome c oxidases (e.g., Cox IV) and other oxidase activities (e.g., NADPH oxidase activities, which are especially high in leukemic blasts [15]). After exposure to oligomycin A, the decrease in the OCR was related to the mitochondrial respiration used to generate ATP (from t_20 min_ to t_26 min_). The remaining respiration is composed of both proton leak and nonmitochondrial respiration and is linked to other oxygen-consuming processes independent of ATP production. The concentration of oligomycin A required to completely inhibit ATP synthase was determined using a range of concentrations, and ATP-linked blast respiration was found to be completely inhibited by 2 μM oligomycin A (Appendix A). Then, to estimate the maximum sustainable respiration by the blasts, FCCP or BAM15 (uncouplers) were used (from t_32 min_ to t_51 min_) in two successive injections (Figure 1C or Appendix A, Appendix A). Following exposure to FCCP, oxygen consumption increased because the mitochondrial inner membrane became permeable to protons. The mitochondrial spare respiratory capacity (SRC) was then calculated by subtracting the FCCP-stimulated OCR (maximal OCR) from the basal OCR (Appendix A). The conditions to correctly reach maximal oxygen consumption will be discussed in the section below. Finally, protein complexes I and III of the respiratory chain were inhibited by a mixture of rotenone and antimycin A to determine the nonmitochondrial respiration of the blasts (from t_56 min_ to t_61 min_). The nonmitochondrial OCR was achieved after injection of a 1 μM antimycin A/rotenone mixture (Appendix A).

### 3.3. Optimization of FCCP Concentrations Required to Reach the Maximal OCR

Maximal mitochondrial oxygen consumption is usually obtained by injection of an uncoupling agent (e.g., FCCP) after inhibiting ATP synthase by oligomycin A. Under these conditions, the transfer of electrons from the respiratory chain is uncoupled from the physiological proton gradient. However, Ruas et al. showed that the maximal respiration observed after FCCP treatment is underestimated in oligomycin-treated glioma and prostate cancer cells [16]. Thus, we first measured the maximal OCR from the blasts of 4 patients following exposure to FCCP preceded or not by treatment with oligomycin A (Figure 2A). The maximal OCR was significantly higher in blasts treated with oligomycin/FCCP than in blasts treated with FCCP alone, suggesting that ATP synthase inhibition is required to obtain maximal respiration in AML blasts.

Next, the maximal OCR of the blasts from 20 different AML patients was achieved after injections of FCCP ranging in concentration from 0.27 to 2.2 µM using XFe24 Seahorse (Figure 2B). The maximal OCR is identified with a black arrow in the upper panel of Figure 2B. Representative OCR profiles (with the blasts from patient #1, #5, #8 and #11) are shown in Appendix A. After exposure to 0.27 µM FCCP, the OCR increased slightly in all samples. However, greater levels were observed in most samples exposed to higher concentrations of FCCP. The maximal OCR (the highest of the 2 OCR values after the addition of FCCP) was obtained with 0.55 µM FCCP for the blasts from patient #8, with 1.1 µM FCCP for the blasts from patients #1, #2, #3, #4, #5, #6, #7, #9, #10, #11, #16, and #20, and with 2.2 µM FCCP for the blasts from patients #12, #13, #14, #15, #17, #18, and #19. These data from 20 different patients showed that the FCCP concentration required to reach the maximal OCR varied depending on the sample analyzed. At the lowest concentrations, the maximum OCR may be underestimated (for instance, see samples #11 exposed to 0.27 or 0.55µM in Appendix A). Conversely, for excessively high concentrations of FCCP, the OCR may collapse (for instances, see samples #1, #5 or #8 exposed to 2µM in Appendix A). This undesirable effect of high concentrations of FCCP is due to the excessive accumulation of this uncoupler in the inner mitochondrial membrane, leading to inhibition of the respiratory activity (Appendix A). Other uncouplers can also be used, such as BAM15 [17]. We successfully tested the BAM15 uncoupler on blast samples and found that BAM15 exhibited a profile similar to that of the FCCP-treated blasts (Appendix A). As previously seen with FCCP, maximal respiration cannot be sustained at the highest concentration of BAM15.

### 3.4. Cytokine Supplementation Is Necessary to Maintain Mitochondrial Metabolism during Culture

Next, we decided to analyze how culture conditions could modify the OCR profiles of blasts. Notably, steps following blood collection inevitably produce cellular stress, which can affect mitochondrial metabolism. Thus, the metabolic changes after blast isolation from two AML patients were evaluated by OCR measurements after being cultured under three different conditions: for 3 h in full RPMI medium, for 18 h in full RPMI medium, and for 18 h in full RPMI medium supplemented with cytokines (Figure 3) (see the composition and concentrations in the Materials and Methods section). The viabilities of the blasts as observed by trypan blue exclusion were greater than 90% after 3 h in full RPMI medium but decreased considerably after 18 h (Figure 3A). However, supplementing the medium with cytokines allowed the blasts to maintain greater than 80% viability for 18 h. Moreover, the OCR of the blasts incubated for 18 h with cytokines did not show a significant change in the OCR profile compared to freshly isolated blasts (within 3 h in full RPMI) (Figure 3B). In contrast, blasts maintained for 18 h in full RPMI (without cytokine supplementation) exhibited a great decrease in mitochondrial respiration, suggesting severe metabolic stress (Figure 3B).

Blast cryopreservation is a method used for the long-term storage of leukemia cells and is applied during most clinical trials to allow retrospective studies to be performed. Therefore, we evaluated the effect of cytokine supplementation on the survival and mitochondrial metabolic organization of thawed blasts after cryopreservation in liquid nitrogen. Blasts were cryopreserved for 45 days (patient #21) or 120 days (patients #26 and #27). Thawed blasts that were cultured for 18 h in full RPMI supplemented with cytokines maintained their viability (>80%) (Figure 4A), preserved their mitochondrial inner membrane potential (for nonapoptotic blasts) (Figure 4B and Appendix A) and exhibited a higher OCR than thawed blasts cultured for 18 h in full RPMI alone (Figure 4C). These results confirm that cytokine supplementation sustains metabolic activity after cryopreservation.

### 3.5. Cryopreservation Slightly Alters Mitochondrial Metabolism of Blasts

We also compared the OCR profiles of freshly collected blasts and thawed blasts after both were cultured for 18 h in full RPMI supplemented with cytokines (Figure 5A–F). In both groups, oximetry analysis showed that 3/5 samples were not statistically different with respect to their basal OCR, ATP turnover, proton leakage or nonmitochondrial OCR. For all 5 samples, the SRC was not different between the freshly collected blasts and corresponding thawed blasts, suggesting the robustness of this metabolic parameter. These results confirm that blasts can be stored for months and will retain a mitochondrial metabolism consistent with that of freshly isolated blasts.

### 3.6. Assessment of OXPHOS in Blasts from Bone Marrow

Finally, we decided to apply our standardized oximetry protocol to measure the OCR from bone marrow blasts isolated from five different patients (Figure 6). The determined OCRs confirmed that this protocol can be easily applied to both bone marrow and peripheral blood blasts. The bone marrow blasts showed slight differences in their basal OCR, maximal OCR and SRC when compared with the peripheral blood blasts from the same patient (Figure 6A–F).

## 4. Discussion

To date, the study of OXPHOS represents a major interest to understand not only the mitochondrial dependency of AML cells [12,18,19,20,21,22] but also their adaptive nongenetic mechanisms of resistance to anticancer treatments such as intensive chemotherapy [13,23,24] or the newly approved targeted therapies IDHi and venetoclax [25,26,27,28,29,30]. In this context, the assessment of a standardized, easy, fast, and robust protocol to measure mitochondrial oxygen consumption in blasts is an urgent need to define whether OXPHOS activity might be a functional biomarker that is predictive of drug response as proposed in Bosc et al., in press [30] and support future clinical trials for AML management.

First, we established and validated a protocol achievable in 6 h to assess mitochondrial respiration in blasts collected from the blood or bone marrow of 20 different AML patients. The parameters for OCR measurements with the XFe24 Seahorse and XFe96 Seahorse, the concentrations of inhibitors and the plates organization have been described in this study (Figure 7). Of note, our recommendation is to use 0.9 × 10^5^ AML blasts per well for XFe96 or 0.25 × 10^6^ blasts per well for XFe24. Moreover, we recommend performing analysis using several technical replicates (at least *n* = 7). This analysis should be performed with blood or bone marrow at the time of diagnosis or relapse to collect enough blasts. Unfortunately, this method will not be reliable to follow the mitochondrial oxidative metabolism of AML blasts in the context of minimal residual disease. In this study, we investigated the metabolism of blasts from peripheral blood mononuclear cells of AML patients. However, it would also be possible to enrich the sample with blasts or with low-frequency subpopulation (as leukemic stem cells) after cell sorting or immunomagnetic isolation [31].

Further, we have shown that OCR assessments can be performed within 3 h following blood draw and up to 18 h if the blasts are maintained in full RPMI medium supplemented with cytokines. Cytokine supplementation is also required to measure the oxygen consumption of thawed blasts from cryopreserved samples. We proposed a culture medium including different cytokines because the heterogeneity of AML increases the difficulty of selective cytokine identification required for the survival of blasts from different patients [32]. Interestingly, we have shown that cytokine supplementation does not seem to erase metabolic differences between blasts since we observed different mitochondrial organizations in most of the blast samples.

Next, this study also underlies that ideal concentration of FCCP requested to measure the SRC is obtained after exposure to increasing FCCP concentrations for each patient sample. Analysis using a single dose of FCCP should be avoided because low FCCP concentration lead to the underestimation of maximal OCR and, conversely, an excess concentration could cause OCR collapse through toxicity. We have observed the same issue with BAM15. Using the SRC to study mitochondrial function in blasts could offer several advantages. The SRC is one of the most robust parameters to determine mitochondrial activity, and its reliability lies in the fact that the SRC is reproducible and very sensitive for the determination of mitochondrial energetic adaptation capacity [33]. Importantly, Schimmer and collaborators showed that AML cells display a lower SRC than normal hematopoietic cells and suggested that this metabolic behavior could be targeted to eradicate AML cells [34].

Recently, gene signatures associated/related to mitochondrial metabolism, OXPHOS, and mitochondrial structures were identified by RNA sequencing. These signatures are predictive of patient response to cytarabine in patient-derived xenograft (PDX) models [23,35] and shorter overall survival of patients from several transcriptomic databases [36]. However, mitochondrial gene signatures cannot fully reflect the dynamics of mitochondrial respiration and the redox homeostasis after various biological stresses. Monitoring mitochondrial oxygen consumption with mitochondrial gene signatures could further improve the prediction of drug response in AML patients, especially in clinical trials.

## 5. Conclusions

These protocols provided here can be used in future clinical studies to evaluate OXPHOS as a biomarker. Monitoring mitochondrial oxygen consumption of blasts could further improve the prediction of drug response in AML patients, especially for drugs targeting mitochondria.

## Figures and Tables

**Figure 1 cancers-13-06353-f001:**
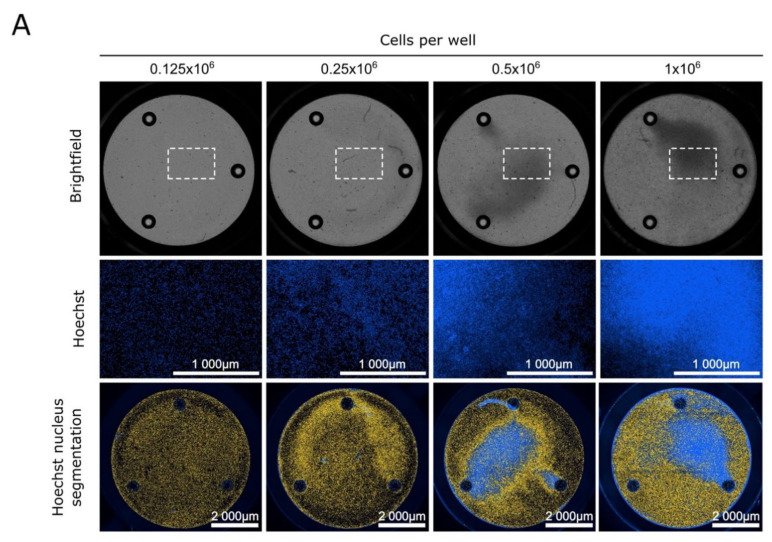
AML cell numbers required to measure oxygen consumption with the XFe24 Seahorse. (**A**) Top panel: Cellular imaging of AML blasts (0.125 × 106 to 1 × 106 cells). Images display the wells of XFe24 cell plates obtained by brightfield light microscopy. Middle panel: Nuclei were stained using a Hoechst probe and visualized with a Cytation I fluorescence cell imager using a DAPI filter (scale bar = 1000 µm). The dotted rectangles represent the selected area of each well. Lower panel: Images obtained after Hoechst staining analyzed using Gen 5 software; the segmentations of the nuclei are shown in yellow (scale bar = 2000 µm). (**B**) Oxygen levels (mmHg) measured in the medium surrounding the AML blasts according to the number of cells per well. At the times indicated (see black arrows), the following drugs were injected: oligomycin A (Oligo; 2 µM), FCCP1 (1.1 µM), FCCP2 (2.2 µM), and antimycin A and rotenone (AA/Rot; 1 µM each). Red dots represent the oxygen levels measured in the wells containing only medium (used for background correction). (**C**) Oxygen consumption rate (OCR; pmol.min^−1^) of blasts from AML patients according to the number of cells per well. Data are the means ± SEM (at least *n* = 3 wells per group). All experiments were performed with blasts freshly collected from the blood of patient #16.

**Figure 2 cancers-13-06353-f002:**
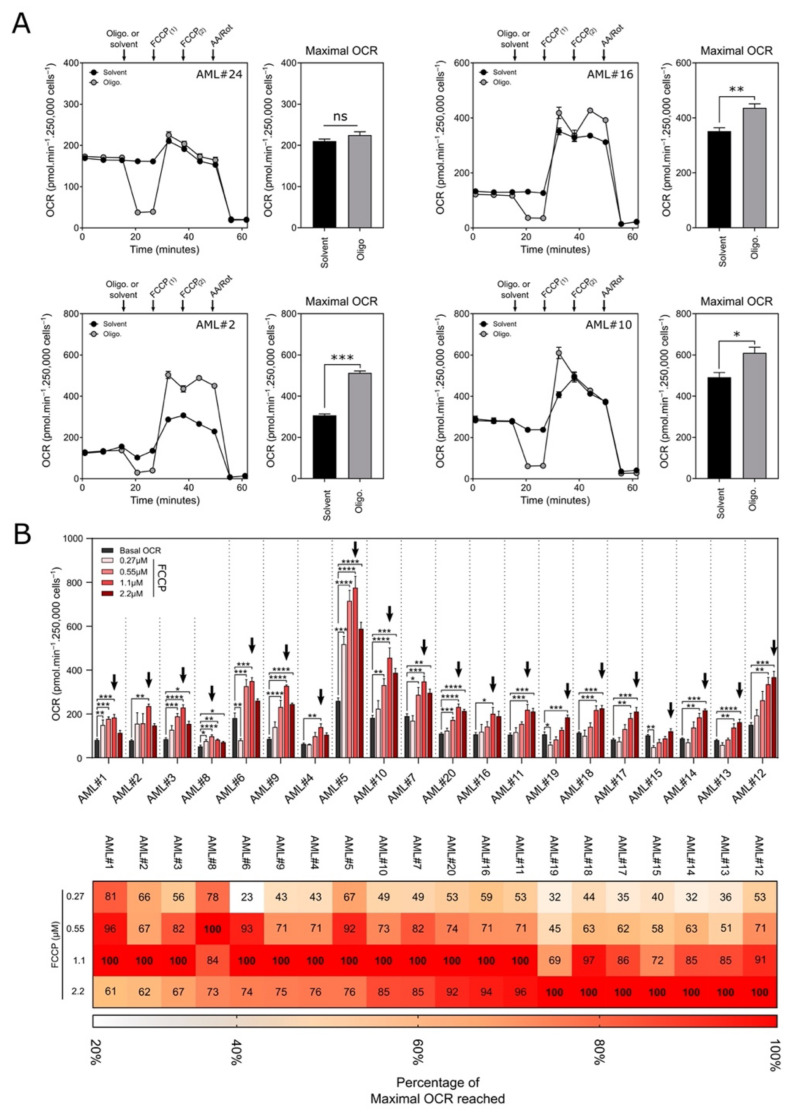
Range of FCCP concentrations required to achieve maximal oxygen consumption in AML blasts. (**A**) Oxygen consumption rate (OCR; pmol∙min^−1^∙250,000 cells^−1^) of blasts from AML patients. After 20 min, 2 µM oligomycin (gray) or medium (black) was injected as indicated, followed by exposure to FCCP1 (1.1 µM), FCCP2 (2.2 µM) and antimycin A and rotenone (AA/Rot; 1 µM each). Experiments were performed with the blasts from the patients indicated in the upper right corner of the OCR profile. The histograms represent the maximal OCR, which corresponds to the highest OCR value measured during FCCP treatment for each condition. (**B**) Top panel-Oxygen consumption rate (OCR; pmol∙min^−1^∙250,000 cells^−1^) of the blasts from 20 AML patients according to the concentration of FCCP. Basal OCRs were measured before oligomycin A injection. Black arrows indicate the highest OCR value obtained after FCCP injection (maximal OCR). Data are the means ± SEM (at least *n* = 3 wells per group). Statistical analyses were conducted with Dunnett’s unidirectional ANOVA multiple comparison test (* *p* < 0.05; ** *p* < 0.01; *** *p* < 0.001; **** *p* < 0.0001). Bottom panel-Heat map of the OCR values obtained according to FCCP concentration. Data are expressed as a percentage of the maximal OCR.

**Figure 3 cancers-13-06353-f003:**
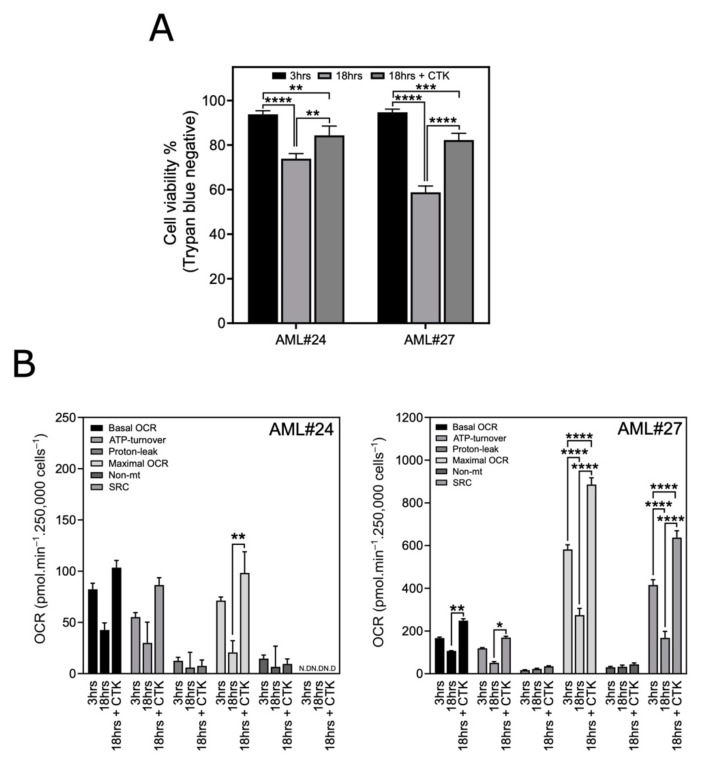
Assessment of the viabilities and OXPHOS parameters of blasts freshly collected from blood and cultured in RPMI medium. Cells were cultured in full RPMI medium for 3 h/18 h or in full RPMI medium supplemented with cytokines (CTK) for 18 h (see the Materials and methods section for the composition and concentrations of cytokines). Experiments were performed with blasts from AML patients #24 and #27. (**A**) Cell viability was determined by trypan blue exclusion. Data are the means ± SD (*n* = 3). (**B**) OXPHOS parameters of blasts cultured as indicated (basal OCR, ATP turnover, proton leak, maximal OCR, non-Mt (nonmitochondrial respiration) and SRC (spare reserve capacity)). Data are the means ± SEM (at least *n* = 3 wells per group). * *p* < 0.05; ** *p* < 0.01; *** *p* < 0.005; **** *p* < 0.0001.

**Figure 4 cancers-13-06353-f004:**
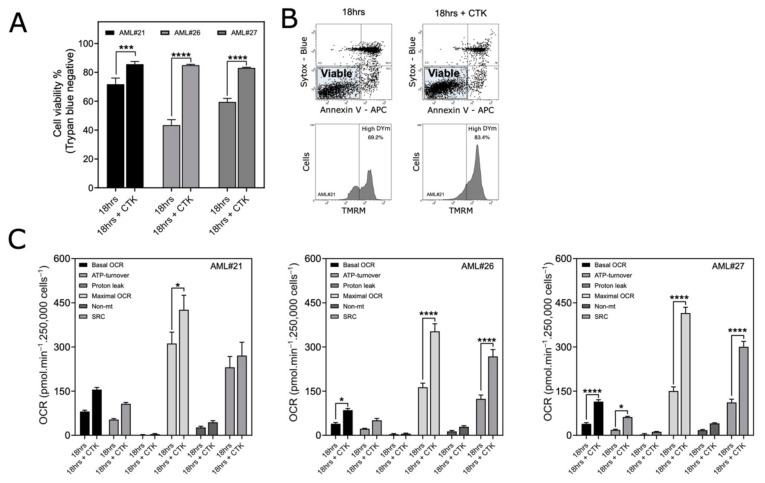
Assessment of the viabilities and OXPHOS parameters of thawed AML blasts cultured for 18 h after cryopreservation in liquid nitrogen. (**A**–**C**) Thawed blasts were cultured in full RPMI (for 18 h) or in full RPMI supplemented with cytokines (18 h + CTK). (**A**) Viability was determined by trypan blue exclusion. Data are the means ± SD (*n* = 3). (**B**) Viable blasts were identified under both conditions by flow cytometry following Annexin V and SYTOX blue staining, and the percentages of blasts with a high mitochondrial membrane potential (ΔΨM) values were determined by TMRM labeling (percentages are indicated in the top right corner of the cytometric profile). The experiment was conducted with blasts from patient #21. (**C**) OXPHOS parameters (basal OCR, ATP turnover, proton leak, maximal OCR, non-Mt (nonmitochondrial respiration) and SRC (spare reserve capacity)) of the thawed blasts cultured as indicated. Data are the means ± SEM (*n* = 3). * *p* < 0.05; *** *p* < 0.005; **** *p* < 0.0001. Experiments were performed with the blasts from the AML patients indicated in the upper right corner of the histograms.

**Figure 5 cancers-13-06353-f005:**
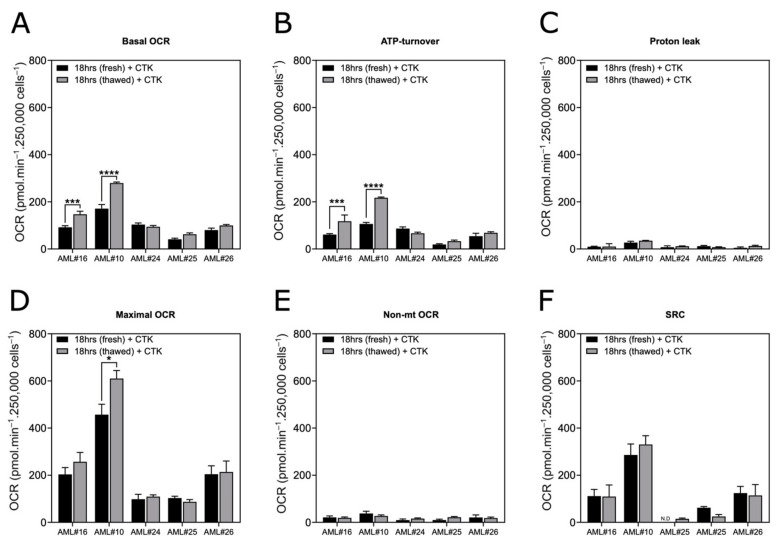
Assessment of the OXPHOS parameters (Basal OCR (**A**), ATP-turnover (**B**), Proton leak (**C**) Maximal OCR (**D**), Nonmitochondrial OCR (**E**), Spare respiratory capacity (**F**)) of freshly collected blasts compared to their corresponding thawed blasts after cryopreservation. All samples were cultivated for 18 h in full RPMI medium supplemented with cytokines. Experiments were performed with the blasts from the AML patients indicated in the histograms. Data are the means ± SEM (at least *n* = 3 wells per group). * *p* < 0.05; *** *p* < 0.001; **** *p* < 0.0001.

**Figure 6 cancers-13-06353-f006:**
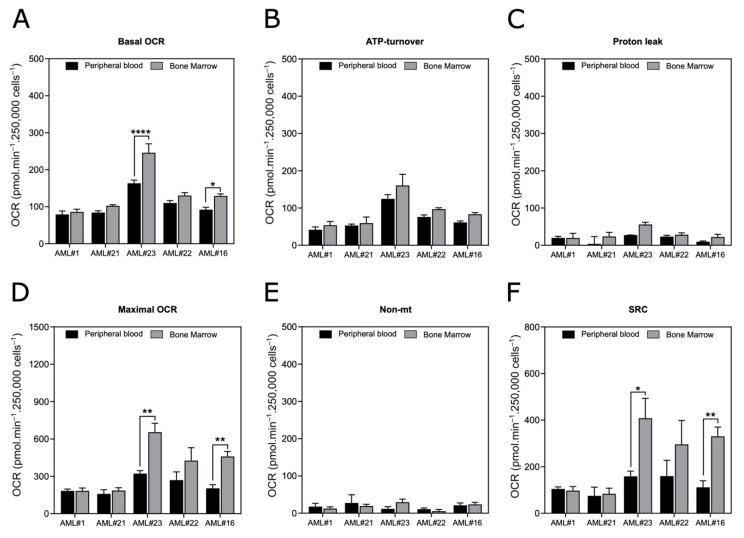
OXPHOS parameters of the blasts isolated from the bone marrow or peripheral blood of AML patients. (**A**–**F**) OXPHOS parameters were measured in freshly isolated blasts from peripheral blood or bone marrow from 5 AML patients. (**A**) Basal OCR, (**B**) ATP turnover, (**C**) proton leak, (**D**) maximal OCR, (**E**) non-Mt (nonmitochondrial respiration) and (**F**) SRC (spare reserve capacity). All samples were cultivated for 18 h in full RPMI medium supplemented with cytokines. Data are the means ± SEM (at least *n* = 3 wells per group). * *p* < 0.05; ** *p* < 0.01; **** *p* < 0.0001.

**Figure 7 cancers-13-06353-f007:**
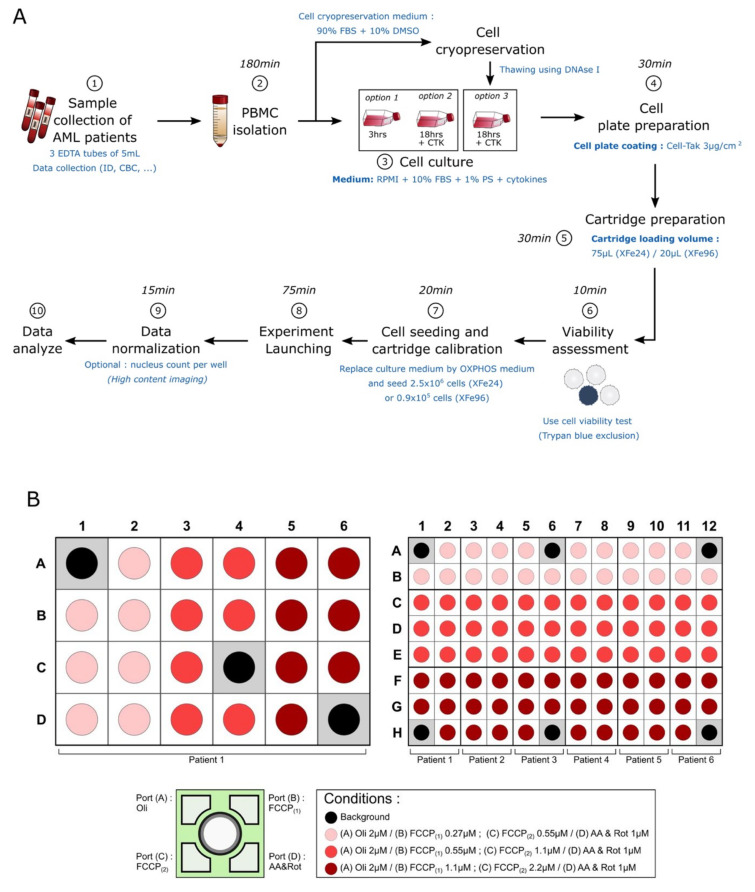
Schematic representation of the standardized protocol for evaluating OXPHOS in AML blasts. (**A**) Workflow representing each step required to analyze the OXPHOS of AML blasts with XFe24 or XFe96 Seahorse. The time required to complete each step is indicated. (**B**) Template organization of the plates used to analyze the OXPHOS parameters of the blasts from one AML patient with XFe24 Seahorse or for 6 AML patients with XFe96 Seahorse.

**Table 1 cancers-13-06353-t001:** Patient characteristics including sex, age, the French-American-British (FAB) classification of AML and the European LeukemiaNet (ELN) risk classification: favorable (1), intermediate (2) or adverse risk (3), s-AML: secondary AML, N.D.: No Data and NA: Not Applicable.

Patient	Sex	Age	FAB	ELN	Cytogenetic Karyotype
AML#1	F	88	M5a	3	47, XX, +6, +8, del(9)(q21), −12, +13, del(17)(p11)
AML#2	M	65	M2	N.D.	46, XY, t(3;3)(q21;q26.2), der(16) t(1;16)(q21;q24)
AML#3	F	75	M1	2	46, XX, add(9)(q3?)
AML#4	M	69	M0	3	46XY, Isochomosomy 11, del(17), t(17,21)
AML##5	M	78	M2	3	45, XY, t(3;3)(q21;q26), −7
AML#6	M	65	N.D.	3	46,XY,del(7)(q12q36)/47,XY,+21/46,XY
AML#7	M	70	M1	2	46,XY
AML#8	M	50	M1	2	46,XY
AML#9	M	71	M5a	3	47,XY,t(1;14)(p32;q32),del(7)(q22q34),der(7)t(7;11)(q34;q22),+13
AML#10	F	49	N.D.	N.D.	N.D.
AML#11	M	38	s-AML	NA	N.D.
AML#12	F	46	M2	1	46,XX,t(8;21)(q22;q22)/45,sl,-X,del(9)(q22)
AML#13	M	54	M1	1	46,XY
AML#14	F	67	M1	2	46,XX
AML#15	F	58	N.D.	2	46,XX
AML#16	M	34	N.D.	3	45,XY,−7,−12,+mar
AML#17	F	86	M4	3	44,X,-X,del(4)(q21),−8,add(9)(p24),add(11)(p15),−16,−17,−12,+19,−20,add(21)(p13),+mar
AML#18	F	67	N.D.	2	46,XX,i(7)(p10)/46,XX
AML#19	M	69	M4	3	46,XY,t(7;21), RUNX1
AML#20	F	86	N.D.	3	46,XX,−6,+8, del(7p) add17p
AML#21	M	30	N.D.	N.D.	N.D.
AML#22	M	67	M3	1	46,XY,t(15;17)(q24;q21)/46,XY, PML-RARα
AML#23	M	59	M4	2	46,XY, EVI1 overexpression
AML#24	F	90	N.D.	N.D.	46,XX
AML#25	M	72	M4	1	46,XY,inv(16)(p13q22)
AML#26	F	81	M4	2	46,XX
AML#27	F	63	M5a	3	48,XX,+8,t(9;11)(p22;q23),+ider(9)(p10).ish t(9;11)(3’KMT2A+;5’KMT2A+),ider(9)(3’KMT2Ax2), KMT2A-MLLT3, EVI1 overexpression

**Table 2 cancers-13-06353-t002:** XFe24 Settings for mitochondrial OXPHOS measurement.

Settings	Cycles	Mix	Wait	Measure	Total Duration
Basal	3	2 min 40 s	2 min	2 min	20 min
Oligomycin (2 µM)	2	1 min 40 s	2 min	2 min	11 min 20 s
FCCP_1_ (0.27–1.1 µM) or BAM15 (0.06–2 µM)	2	1 min 40 s	2 min	2 min	11 min 20 s
FCCP_2_ (0.55–2.2 µM) or BAM15 (0.13–2.5 µM)	2	1 min 40 s	2 min	2 min	11 min 20 s
Antimycin A + Rotenone (1 µM each)	2	1 min 40 s	1 min 20 s	2 min	10 min

## Data Availability

The data presented in this study are available in this article.

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
