# Peer review of "Clinically Relevant Oxygraphic Assay to Assess Mitochondrial Energy Metabolism in Acute Myeloid Leukemia Patients"

_cancers, 2021, doi:10.3390/cancers13246353_

Round 1

Reviewer 1 Report

Summary

Fovez et al. is a manuscript that presents a standardized method to assess the metabolic activity of both peripheral and bone marrow-derived AML blast cells using the Agilent Seahorse Extracellular Flux analyzers. The authors report the methods for cell preparation, preparation of the analysis plates, and compound concentrations. The authors convincingly demonstrate through their data the ideal cell numbers and attachment procedures for the cells, inhibitor concentrations, the correct order of inhibitor addition and times for analysis. All in all, the authors have presented a method that others can use as a standardized approach to assessing mitochondrial metabolism as a biomarker in AML blasts.

Brief Comments

Generally, the article is well written and supported by suitable references and data. The authors have done a commendable job in most aspects of the article. However, the authors should consider the following points that apply throughout the manuscript to improve it:

  1. Include an analysis of the data and examples that would aid a non-expert reader. The authors should be reminded that for this to be used as a standardized method, scientists who are not experienced in metabolism may refer to the article as a resource. Therefore, in some instances of the rationale for choices should be provided or data further explained.
  2. Specificity as it relates to instruments and compounds. At instances in the manuscript, the authors present alternatives. However, there is not a thorough discussion of these, and again the non-expert may not be exactly clear from the article if they can be suitably interchanged or not (protonophore and CellTak). The authors should pay more attention to this point.
  3. Improvement of figures and tables. Some instances exist where the authors could improve either the illustrations or clarify information presented in tables.
  4. The authors should state if the eight well format of the XF extracellular flux analyzer can be used, or if they think it does not present enough robustness for repetition, state that as well. I believe this is important in setting the reader's expectations in how they can reliably apply this method.

All these points are minor and should not hinder the swift revision of the manuscript.

Specific Comments

Below are specific instances that expand on the general comments provided above.

  • Line 48- as it relates to the ELN, this sentence should perhaps be reworked for accuracy. A suggestion would be "…to provide prognostic information and allow for the stratification of patients into three risk categories (favorable, intermediate or adverse)". Currently, the wording of the text there seems to not fully capture the ELN points.
  • Line 54- for reference #5, several other references could be included here that are recent and either partially or fully support this point. For instance, Sofie et al. (10.3390/cells9051155) & Panina et al. (10.1186/s40170-021-00253-w).
  • Line 64- Here, OXPHOS is simply stated in the text; presumably, OXPHOS activity or levels or something similar was intended, as OXPHOS alone does not address what the biomarker would be.
  • Line 71- For completeness, should the XF 8 well version not be included, which is the 8 well version of the instrument's larger 24 and 96-well formats?
  • Line 90- Should the specific IRB number granted for this work not be included here- this can be compared against the journal's requirements.
  • Line 220- Notably, there is no Fig 1C in the supplemental material. Perhaps this was meant to be one of the panels, but the authors should correct this labelling issue. Additionally, the use of ionophores is acceptable, but no other protophore has been used so far, so I think it might be best to say FCCP or state FCCP and B15 as might be the authors' intention.
  • Line 238- It seems more logical to have the inhibitors listed as oligomycin/FCCP to reflect this the order that the compounds were added to the cells. Of note, this experiment is to be commended and demonstrates careful thought on the part of the authors.
  • Line 266- The authors could provide some patient cell numbers for the collapse with excess FCCP, e.g. #1, #2, #3, #6, among others
  • Line 269-270- This is true and a worthwhile point; one thing to note is that the BAM15 maximal respiration cannot be sustained at the highest concentration. This is something that the authors in the text could highlight. Additionally, the authors should provide in their table the concentration profile of BAM15. Otherwise, it is correct, and for #16, the profile follows that of FCCP.
  • Line 275- the statement' mitochondrial metabolic organization' is odd and mitochondrial metabolism here would seem adequate.
  • Line 306- The phrase 'a high level of energy' is not 100% accurate, and 'sustains metabolic activity' or similar seems more accurate.
  • Line 319- Rewording the sentence to put the slightly before the verb might allow for a better flow
  • Line 327- 'consistent with' might be a better wording than 'close to'.
  • Line 380- The phrase 'using a single dose of uncoupler should be avoided' should be expanded on more. The reasoning for this should be clearly stated for the inexperienced reader who may be using this only as a resource. It is worthwhile to point out the ideal concentration can be easily missed and instead lead to toxicity.

Figures & tables

  • Table 1- similar to s-AML, the abbreviation for N.D.- no data and NA- not applicable should be stated in the legend above the table.
  • Table 2 in the manuscript does not appear any different to the Supplemental table 1 provided. This is indeed curious as the authors point out that the procedure for the 96 well format is provided in Suppl. Table 1, the authors should check this table and if no differences exist in the protocol, point this out to the reader in the text.
  • Additionally, Table 2 should include the concentrations of the inhibitors. Inclusion here would seem like a more logical location and assist the readers who wish to use this article as a resource, as they can quickly glance at the table for details of concentration.
  • Figure 1- the individual panels should either be labelled here or the authors should consider increasing the size of the font and legend for the cell numbers. This is such that the reader can quickly identify how each of these panels of data are different. If this cannot be done, then the authors should consider using letters for each panel.
  • Supplemental Figure S4- Indeed, it is good that #14 is seen as a generally representative trace in the supplemental data, BUT it would also be worthwhile to see a trace for one of the higher maximal respiratory capacity patient cells, e.g. #5, #10 or #12. Or cells like #8, where there is minimal increase and there is FCCP toxicity as the concentration increases (perhaps #2). This allows the reader to observe the actual oxygen consumption vs time and observe if each cell line had a two-step increase to maximal as is the case in #14 or if some cell lines increase immediately to maximal. Additionally, the authors should offer some brief comments on these results so that the reader who is using the paper as a resource knows that such results are possible.
  • Figure 6- This reviewer is unclear what the statistical analysis here shows, other than significant differences between patients. This is notably a point that is made nowhere else in the manuscript and appears oddly here.
  • Discussion- There seems to be a parenthesis before each of the numbered references in the square brackets; the authors should correct

Author Response

Brief Comments

"Generally, the article is well written and supported by suitable references and data. The authors have done a commendable job in most aspects of the article. "

Answer: We thank the reviewer for his comments in our paper.

"However, the authors should consider the following points that apply throughout the manuscript to improve it:Include an analysis of the data and examples that would aid a non-expert reader. The authors should be reminded that for this to be used as a standardized method, scientists who are not experienced in metabolism may refer to the article as a resource. Therefore, in some instances of the rationale for choices should be provided or data further explained."

The manuscript is now clarified for non-expert reader. We have improved the manuscript (see highlighted version and comments below) answering to all comments of the reviewer. We are thankful to the reviewer for this positive and thorough evaluation.

"Specificity as it relates to instruments and compounds. At instances in the manuscript, the authors present alternatives. However, there is not a thorough discussion of these, and again the non-expert may not be exactly clear from the article if they can be suitably interchanged or not (protonophore and CellTak). The authors should pay more attention to this point."

Answer: Polylysine is slightly less effective than Cell tak, specially with blasts from patients. So we decided to remove poly-Lysine from the manuscript to avoid confusion. We have also improved the manuscript with more explanations about collapse of OCR with high concentrations of BAM15 or FCCP (see comments below). Supplementary Fig 4 is modified to clearly visualize the collapse of OCR in samples.

"Improvement of figures and tables. Some instances exist where the authors could improve either the illustrations or clarify information presented in tables.-"

Answer: Figure 1, Figure 6, Supplementary Fig.1, Supplementary Fig 4,  table 1, table 2 and supplementary table 1 have been modified following reviewer recommendations (see comments below)

   "The authors should state if the eight well format of the XF extracellular flux analyzer can be used, or if they think it does not present enough robustness for repetition, state that as well. I believe this is important in setting the reader's expectations in how they can reliably apply this method."
Answer: We agree with reviewer, but we don't have eight wells format XF extracellular analyzer to compare with our data from XFe24 and XF96e Seahorse.  However, we have modified the manuscript (line 363) with the sentence "we recommend performing analysis using several technical replicates (at least n=7)".

Specific Comments

Line 48- as it relates to the ELN, this sentence should perhaps be reworked for accuracy. A suggestion would be "…to provide prognostic information and allow for the stratification of patients into three risk categories (favorable, intermediate or adverse)". Currently, the wording of the text there seems to not fully capture the ELN points.
Answer: We agree with reviewer, and we changed the sentences.

Line 54- for reference #5, several other references could be included here that are recent and either partially or fully support this point. For instance, Sofie et al. (10.3390/cells9051155) & Panina et al. (10.1186/s40170-021-00253-w).
Answer: We agree with reviewer, both references are added.

Line 64- Here, OXPHOS is simply stated in the text; presumably, OXPHOS activity or levels or something similar was intended, as OXPHOS alone does not address what the biomarker would be.
Answer: We agree with reviewer, and we changed « OXPHOS » to « mitochondrial oxygen consumption » in the revised manuscript.

Line 71- For completeness, should the XF 8 well version not be included, which is the 8 well version of the instrument's larger 24 and 96-well formats?Answer: Unfortunately, our laboratory does not have a XF Seahorse with 8 wells. We think this version is not adapted to this kind of study. As seen in Fig. 7, we propose to measure OXPHOS in blast samples using at least 7 technical replicates. A sentence line 364 is added to give recommendation about technical replicate: " Moreover, we recommend performing analysis using several technical replicates (at least n=7)."

Line 90- Should the specific IRB number granted for this work not be included here- this can be compared against the journal's requirements.
Answer: As required, the specific number is added line 423. Moreover, as requested by the editor, we sent a blank form of the consent for publication.

Line 220- Notably, there is no Fig 1C in the supplemental material. Perhaps this was meant to be one of the panels, but the authors should correct this labelling issue. Additionally, the use of ionophores is acceptable, but no other protophore has been used so far, so I think it might be best to say FCCP or state FCCP and B15 as might be the authors' intention.
Answer: As required, the missing label "C" is added, and legend is modified. Moreover, we increased the size of the fonts for legends included in the pictures (see modified supplementary Figure 1). In the manuscript, we have also removed the term "protonophore" line 220 and added "BAM15".

Line 238- It seems more logical to have the inhibitors listed as oligomycin/FCCP to reflect this the order that the compounds were added to the cells. Of note, this experiment is to be commended and demonstrates careful thought on the part of the authors.
Answer: We agree with reviewer, we changed the sentences.

Line 266- The authors could provide some patient cell numbers for the collapse with excess FCCP, e.g. #1, #2, #3, #6, among others`

Answer: We agree with reviewer, we have completed supplementary Fig S4 with data to show the collapse of OCR with excess FCCP. We added the following sentence from line 265: At the lowest concentrations, the maximum OCR may be underestimated (for instance, see sample #11 exposed to 0.27 or 0.55µM in Fig. S4). Conversely, for excessively high concentrations of FCCP, the OCR may collapse (for instances, see samples #1, #5 or #8 exposed to 2µM in Fig. S4).

Line 269-270- This is true and a worthwhile point; one thing to note is that the BAM15 maximal respiration cannot be sustained at the highest concentration. This is something that the authors in the text could highlight. Additionally, the authors should provide in their table the concentration profile of BAM15. Otherwise, it is correct, and for #16, the profile follows that of FCCP.
Answer: As required, we added the range of concentrations of BAM15 in table (see modified table 2 and supplementary table 1).  A sentence is added line 271 to highlight the collapse of maximal respiration at the highest concentration of BAM15: " As previously seen with FCCP, maximal respiration cannot be sustained at the highest concentration of BAM15". 

Line 275- the statement' mitochondrial metabolic organization' is odd and mitochondrial metabolism here would seem adequate.
We agree with reviewer, and we changed the sentences. The same correction is done line 326/327.

Line 306- The phrase 'a high level of energy' is not 100% accurate, and 'sustains metabolic activity' or similar seems more accurate.
We agree with reviewer and we changed the sentences.

Line 319- Rewording the sentence to put the slightly before the verb might allow for a better flow
We agree with reviewer and we changed the sentences.

Line 327- 'consistent with' might be a better wording than 'close to'.
We agree with reviewer and we changed the sentences.

Line 380- The phrase 'using a single dose of uncoupler should be avoided' should be expanded on more. The reasoning for this should be clearly stated for the inexperienced reader who may be using this only as a resource. It is worthwhile to point out the ideal concentration can be easily missed and instead lead to toxicity.
Answer: We agree with reviewer, we changed the sentences to clarify the discussion (line 381-l385).

Figures & tables

Table 1- similar to s-AML, the abbreviation for N.D.- no data and NA- not applicable should be stated in the legend above the table.
Answer: As suggested, we added these information in the legend.

Table 2 in the manuscript does not appear any different to the Supplemental table 1 provided. This is indeed curious as the authors point out that the procedure for the 96 well format is provided in Suppl. Table 1, the authors should check this table and if no differences exist in the protocol, point this out to the reader in the text.
Answer: We apologize for this mistake. Table 2 has some errors; some data are now corrected.

Additionally, Table 2 should include the concentrations of the inhibitors. Inclusion here would seem like a more logical location and assist the readers who wish to use this article as a resource, as they can quickly glance at the table for details of concentration.
Answer: As required, we added all the concentrations of inhibitors, including also BAM15 (see modified table 2 and supplementary table 1).

Figure 1- the individual panels should either be labelled here or the authors should consider increasing the size of the font and legend for the cell numbers. This is such that the reader can quickly identify how each of these panels of data are different. If this cannot be done, then the authors should consider using letters for each panel.
Answer: As required, we increased the size of the fonts for legend included in the panel (see modified Figure 1 and modified supplementary Fig S1).

Supplemental Figure S4- Indeed, it is good that #14 is seen as a generally representative trace in the supplemental data, BUT it would also be worthwhile to see a trace for one of the higher maximal respiratory capacity patient cells, e.g. #5, #10 or #12. Or cells like #8, where there is minimal increase and there is FCCP toxicity as the concentration increases (perhaps #2). This allows the reader to observe the actual oxygen consumption vs time and observe if each cell line had a two-step increase to maximal as is the case in #14 or if some cell lines increase immediately to maximal. Additionally, the authors should offer some brief comments on these results so that the reader who is using the paper as a resource knows that such results are possible.
Answer: We agree with reviewer, and we have completed supplementary Fig S4 with data to show the collapse of OCR with excess FCCP. The discussion is clarified (line 382-387). Modified supplementary Fig 4 is also now cited from line 265.

Figure 6- This reviewer is unclear what the statistical analysis here shows, other than significant differences between patients. This is notably a point that is made nowhere else in the manuscript and appears oddly here.
Answer: As suggested, we removed the statistical analysis between patients (see new Fig 6).

Discussion- There seems to be a parenthesis before each of the numbered references in the square brackets; the authors should correct
As suggested, we removed the parenthesis.

Reviewer 2 Report

With the increasing focus on metabolic pathways in leukemia, this is a timely and useful technical paper which will be of relevance to anyone working in cancer metabolism.

Overall the paper is comprehensive and well written. I had a couple of very minor comments:

Table 1 should contain a footnote explaining what number 1-3 are, and abbreviations N/A and ND

Section 2.3. It would be worth commenting that settings were same for both XFe24 and Xfe96 (Table 2 and Suppl Table 1).

Author Response

With the increasing focus on metabolic pathways in leukemia, this is a timely and useful technical paper which will be of relevance to anyone working in cancer metabolism.Overall the paper is comprehensive and well written.

Answer: We thank the reviewer for his comments in our paper.

I had a couple of very minor comments:

Table 1 should contain a footnote explaining what number 1-3 are, and abbreviations N/A and ND
Answer: As suggested, we added all these informations in the legend.

Section 2.3. It would be worth commenting that settings were same for both XFe24 and Xfe96 (Table 2 and Suppl Table 1). Answer: We apologize for this mistake. Table 2 has some errors; data are now corrected.

Round 2

Reviewer 1 Report

Very good on the corrections, the manuscript reads well and is now accessible for all levels of readers.